# Long-Term Impact of Transhumance Pastoralism and Associated Disturbances in High-Altitude Forests of Indian Western Himalaya

Shiekh Marifatul Haq [1,2], Umer Yaqoob [1], Eduardo Soares Calixto [3], Manoj Kumar [4], Inayat Ur Rahman [5,*], Abeer Hashem [6], Elsayed Fathi Abd_Allah [7], Maha Abdullah Alakeel [6], Abdulaziz A. Alqarawi [7], Mohnad Abdalla [8], Fayaz A. Lone [9], Muhammad Azhar Khan [5], Uzma Khan [5] and Farhana Ijaz [5,*]

1   Department of Botany, University of Kashmir Srinagar, Srinagar 190006, India; snaryan17@gmail.com (S.M.H.); umerraj6668@gmail.com (U.Y.)
2   Wildlife Crime Control Division, Wildlife Trust of India, Noida 201301, India
3   Institute of Food and Agricultural Sciences, University of Florida, Gainesville, FL 32611, USA; calixtos.edu@gmail.com
4   GIS Centre, IT & GIS Discipline, Forest Research Institute, PO New Forest, Dehradun 248006, India; manojfri@gmail.com
5   Department of Botany, Hazara University, Mansehra 21300, Pakistan; azharfinal@gmail.com (M.A.K.); uzmaqau2003@yahoo.com (U.K.)
6   Botany and Microbiology Department, College of Science, King Saud, University, Riyadh 11451, Saudi Arabia; habeer@ksu.edu.sa (A.H.); 441203162@student.ksu.edu.sa (M.A.A.)
7   Department of Plant Production, College of Food and Agriculture Science, King Saud University, Riyadh 11451, Saudi Arabia; eabdallah@ksu.edu.sa (E.F.A.); alqarwy@ksu.edu.sa (A.A.A.)
8   Key Laboratory of Chemical Biology (Ministry of Education), Department of Pharmaceutics, School of Pharmaceutical Sciences, Cheeloo College of Medicine, Shandong University, 44 Cultural West Road, Jinan 250012, China; mohnadabdalla200@gmail.com
9   Department of Botany, Government Degree College (Women), Kupwara 193222, India; falonebotany@gmail.com
*   Correspondence: hajibotanist@outlook.com (I.U.R.); fbotany@yahoo.com (F.I.)

**Abstract:** The Himalayan Mountains are geodynamical important, featuring a wide climatic range with a rich diversity of flora, fauna, human communities, culture, and social set-up. In recent decades, due to constant anthropogenic pressure and considerable changes witnessed in the climate of the region, species of this region are threatened. Here, we assessed the impact of nomadic settlement and associated disturbances on plant species composition, diversity parameters, ecosystem properties, and fire incidence in high-altitude forests of Western Himalaya, India. Based on the distance between nomadic settlement location and forest, we classified forest as near nomadic settlement (NNS) or away nomadic settlement (ANS) forest types. We found a significant variation in plant species composition between forest types. Three species, namely, *Sibbaldia cuneata*, *Poa annua*, and *Abies pindrow*, contribute 25% of the cumulative variation in plant species composition. Studying live plants, we found a significant difference only for density, in which ANS had a higher average density than NNS. Considering dead plants, we found a significant difference in all nine plant-related parameters evaluated between sites. NNS had a higher value of all parameters evaluated, except for height, which was higher in ANS sites. ANS forest type show 1.3 times more average carbon stock (160.39 ± 59.03 MgCha$^{-1}$; mean ± SD) than NNS forest type (120.40 ± 51.74 MgCha$^{-1}$). We found a significant difference in plant diversity evaluated between forest types. ANS had higher values of Margalef and Fisher diversity but lower values of evenness. We found that NSS had significantly higher values of fire incidences, whereas ANS has a higher normalized differential vegetation index and enhanced vegetation index. Overall, our study showed that species composition, diversity, and fire incidence are strongly impacted due to nomadic settlements. These findings are paramount for designing appropriate livelihood options for indigenous communities and management policies of the long-term forest harvest to achieve global goals and the UN Decade on Ecosystem Restoration targets (2021–2030) to protect the sustainable development of forest mountainous regions.

**Keywords:** ecosystem properties; diversity; biosphere; forests; Kashmir Himalaya

## 1. Introduction

The human land-use pattern is increasingly recognised as an important step in interpreting the structure and functional dynamics of ecosystems to understand its ecological implications [1–3]. The anthropogenic activities in forest ecosystems have a profound impact not only on forest structure and composition [4,5], but also on ecological processes [6,7], biodiversity [8], and nutrient availability [9]. Forests play a vital role in maintaining ecosystem stability by absorbing a large chunk of carbon. Forests act as a sink both globally and regionally for storing huge quantities of carbon. This carbon gets released into the atmosphere if deforestation, burning of fuelwood, and other human activities continue unabated [10]. In the last 25 years, carbon stocks have decreased by 11 giga tons (Gt) in the global forest area [11].

Understanding the vegetation and ecosystem features of forests in the region is a prerequisite for describing various ecological processes as well as modelling forest functions and dynamics [12–20]. Forests have to cope in their environment with a multitude of natural and anthropogenic forms of stress. Resilience and resistance mechanisms to biotic and abiotic stresses are of particular importance for long-lived tree species. The degradation of forests is mainly due to various anthropogenic pressures [21–24]. Among all the anthropogenic pressures that the forests are experiencing, the nomadic settlements inhabiting the forests worldwide are one of the major factors affecting the forests [25].

The nomadic settlements and transhumance occurring throughout the world have become major disturbances and affect the resilience and resistance of forest ecosystems. The nomadic settlements have influenced the forests of the Himalayas already for millennia [26,27]; however, the influence of climate change and land-use changes have further aggravated the impact in recent decades [28]. Nutrient enrichment in the neighbourhood of corrals due to deposition of dung and urine results in changes in plant composition and edaphic factors along the grazing gradient [29]. Lange (1969) coined the term "piosphere" to describe this localised impact of disturbance caused by the grazing on vegetation and soils [30]. The nitrogen accumulated in these sites is a source of ammonia, a major cause of invasion by new species [29,31].

The grazing pressure by a limitless dairy cattle population around the nomadic settlements also leads to invasion by several weed plants. These weeds, being exceptionally allelopathic, have out-competed the tree recovery in the whole locale. The indicators of these patches are a constant decrease in biomass, changes in the composition of species, and soil compaction [32]. The rise in the scrub area in these piospheric hotspots reflects the expression of the tremendous grazing pressure known as the Nibble effect [33,34]. In order to accommodate secondary vegetation, the forests are converted to scrub areas and grasslands. Sudden spikes in naked rocks can also be due to endless grazing, since it removes the vegetative cover excessively, speeding the solifluction process, which increases the area under rocks and stones [34].

In Kashmir Himalaya, various anthropogenic activities including deforestation, overgrazing of pastures, agricultural intensive practices, and urbanisation have increased over the past years [35,36]. However, due to constant anthropogenic pressure on high-altitude forests and considerable changes in the ecosystem properties, there is a great threat to species in this region. Satellite-based earth observation data have been widely used to detect disturbed and protected forest covers at landscape, regional, and global scales [14,37–39]. The satellite-based normalized difference vegetation index (NDVI) and enhanced vegetation index (EVI) were used to observe the forest land cover disturbance. Landsat images were used to observe the percentage of forest cover as well as the level of disturbance in the forests [40].

In this study, our main hypothesis is that the forests near nomadic settlements would be more negatively impacted than the forests away from nomadic settlements. To test this,

we evaluate nine plant traits of live and dead plants of forested landscapes existing near and away from nomadic settlements. Similarly, we compare the various indices (species richness, dominance, Shannon, Simpson, Pielou Evenness Index, Margalef Richness Index, equitability, and Fisher alpha) between the sites to visualise the impact. Lastly, we compare the variation of fire incidence, NDVI, and EVI between these two distinguished forest sites.

We show how nomadic settlements and accompanying anthropogenic stress interactions influence the composition, diversity, and ecosystem properties of plants. Such empirical data can help to evolve scientific policy management tools for effectively restoring degraded forest ecosystems of this Himalayan region.

## 2. Materials and Methods

### 2.1. Study Area

The study area (Bangus) falls in the Northern part of Kashmir Valley in district Kupwara of the Western Himalayas (Figure 1). The Bangus Valley is 72 km from Srinagar and stands at 10,000 feet above sea level. Bangus is a combination of two words: *Bun*, forest, and *Gus*, which means grass. The Bangus Valley is located between 34°35′10.26″ and 34°36′56.71″ N latitude and 74°04′22.27″ and 74°04′30.68″ E longitude at an elevation of 1550–3450 masl in Kupwara District, Kashmir Himalaya, India (Figure 1). It is a linear elliptical bowl-shaped valley aligned along the east–west axis and is bounded by Mawar and Rajwar in the east, Dajlungun and Shamasbury Mountains in the west, Chowkibal and Karnah Guli in the north, and Kazinag Range in the south. The region is largely drained by the Kishan Ganga River. Average minimum and maximum temperatures vary from −5 °C to 32 °C. The average annual rainfall in this region is 869 mm. Bangus is endowed with a variety of flora and fauna. The valley is replete with many medicinal plants and 50 species of animals, of which 10 species are birds [41]. The forest of the region is predominantly influenced by the nomadic tribal communities of "Bakarwal" and "Gujjar" tribes. Forests of the study region have permanent huts (locally called as "doke") built by nomadic communities that are revisited usually after the end of winter seasons. During the winter period (start or the end of October to April, based on the snow fall time), these communities leave the forests and move away to stay in downside villages away from the forests. Huts inside the forests are revisited usually each year by these tribes. The forests near the huts can clearly be distinguished from forests away from the huts (Figure 2).

### 2.2. Sampling Design and Measurements

Field sampling and measurement were conducted with the quadrat methods following [36]. Specifically, for each of the selected forest types, we performed systematic random floral sampling, to obtain data on vegetation parameters at twenty-four square-shaped 0.1-hectare (hereafter ha) plots (i.e., 3 plots × 4 sites × 2 forest types = 24 plots). In each plot, we recorded plant species habits, namely, trees, shrubs, and herbs. For tree sampling, we recorded the density of live and dead trees within each plot. Shrubs were sampled in four square shaped subplots (5 m$^2$) delimited within each plot. Finally, herbaceous diversity was sampled in five square shaped (1 m$^2$) subplots, one in each plot's corner and one in the middle. In total, ninety-six (12 subplots × 4 sites × 2 forest types = 96) (5 m$^2$) sub plots for shrubs and one hundred twenty (15 subplots × 4 sites × 2 forest types = 120) (1 m$^2$) sub plots for herbs were sampled in the present study. The tree species composition and DBH (diameter at breast height, i.e., 1.37 m) of each tree were recorded. The tree species richness was simply taken as the total number of species recorded within the plots. The tree biomass and C stock for live and dead trees were calculated following Haq et al. [42]. The biomass and C stock were estimated based on variables such as density, DBH, and height of tree species.

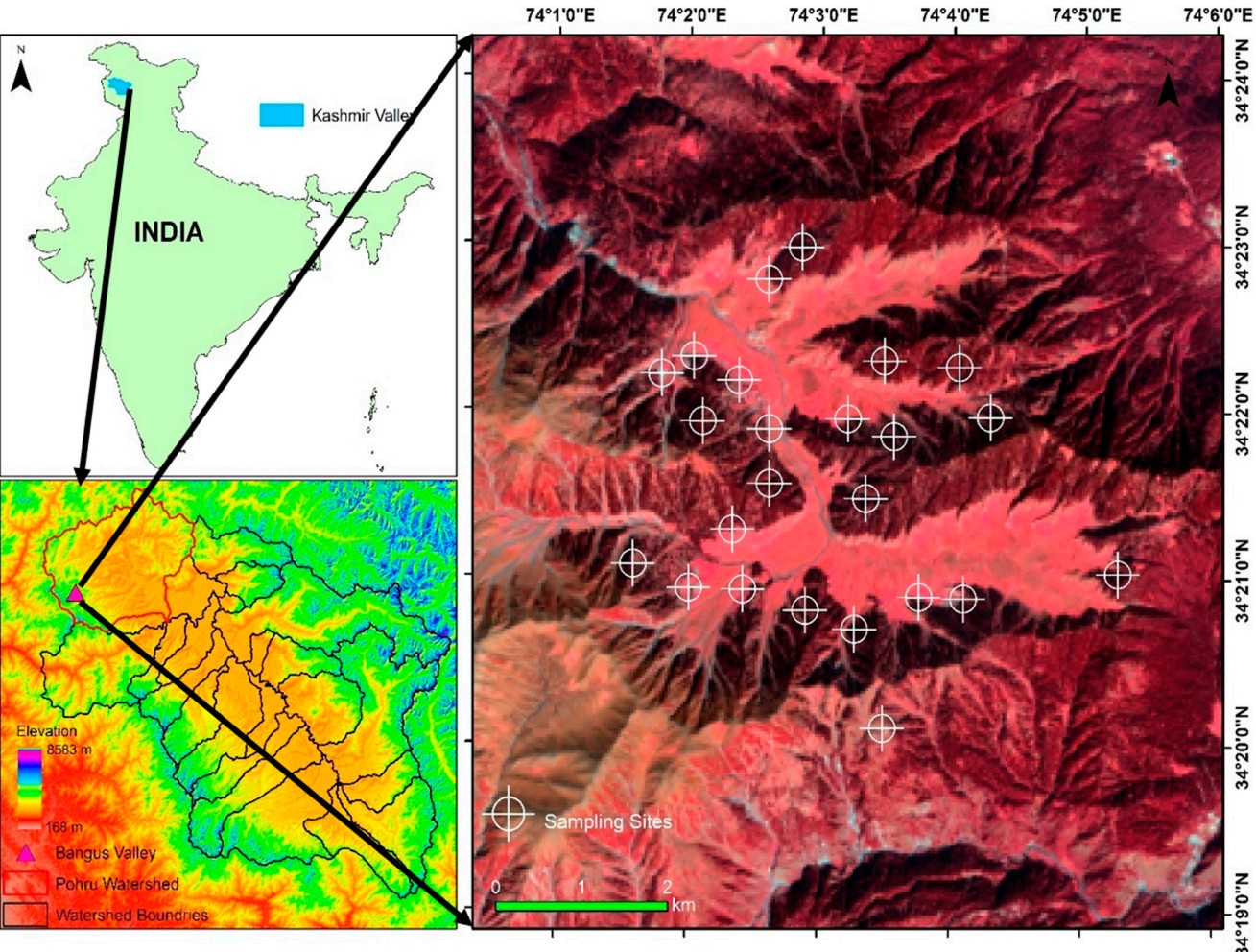

**Figure 1.** Map showing the location and topography of the Bangus Valley and points showing the sampling plots in the study area.

We also measured the relative degree of disturbance in each sampling forest type. Based on the distance between nomadic settlement location and forest, we classified forest as near nomadic settlement (NNS) or away nomadic settlement (ANS) forest types. We measured the distance with the help of measuring tape. Near nomadic settlement, forest types were defined as forests with nomadic settlements within a radius of 200–250 m around nomadic communities where the impact of nomadic settlement and associated disturbances could be visualised, whereas away from nomadic settlement forest types were defined as forests without nomadic settlements around or near sampling plots. The degree of disturbance was measured at each forest site using the semi-quantitative scale [43]. Based on visual assessment at each sampling plot, a three-point scale (0 = none, 1 = moderate, 2 = high) was used to record disturbance levels. The disturbance factors such as, distance from nomadic settlements (DFNS), stem cutting (CT), habitat alteration (HA), forest degradation (FD), tree lopping (TL), soil erosion (SE), forest fire (FF), and livestock grazing (LG) were taken into consideration (Figure 2). A value of 0 indicates that the evaluated disturbance parameter has no impact on the forest type and that nomadic settlements are far away from the forest, whereas a value of 2 indicates that the evaluated disturbance parameter has a high impact on the forest and that human settlements are in close proximity to it. A value of 1 denotes a moderate level of the evaluated disturbance parameters.

The plant specimens were collected from the field to serve as herbarium voucher specimens following standard herbarium techniques [44]. Plant specimens, which remained unidentified in the field, were identified using standard taxonomic tools and literature [45]

(http://www.efloras.org/, accessed on 1 August 2021) and were further authenticated by matching the plant specimens with those deposited in KASH herbarium (https://species. wikimedia.org/wiki/KASH, accessed on 1 August 2021). For the nomenclature update, the Plant List in the online taxonomic database (www.theplantlist.org, accessed on 1 August 2021) was used. With the abundance and number of species from each plot, we calculated the importance value index (henceforth IVI) for each plant species per plot per forest site. The Importance Value Index (IVI) was used to assess the plant species' dominance [43]. Global positioning system (GPS Garmin map76cs) was used to record altitude as well as geo-coordinates of the forest sites [42].

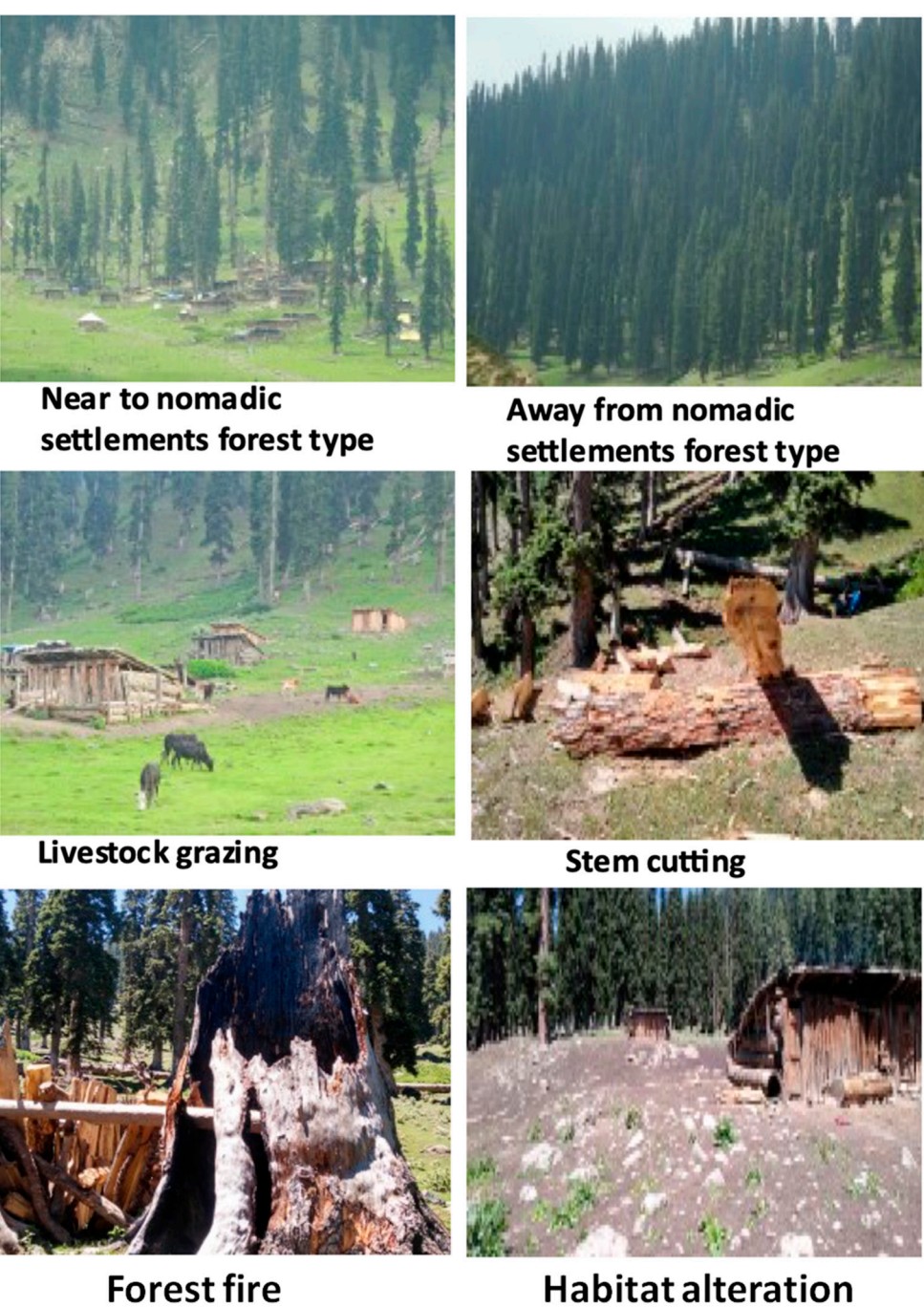

**Figure 2.** Various observations of forests near nomadic settlement (NNS) and away nomadic settlement (ANS) as seen during field visits.

### 2.3. Remote Sensing for Identifying the Highly Disturbed Hotspots

We used satellite-based NDVI and EVI around NNS and ANS sites to evaluate nomadic settlement impacts on the forest land cover disturbance. In this regard, Landsat 8 Operational Land Imager (OLI) collection 1 surface reflectance Level-2 data products at a spatial resolution of 30 meters were procured from U.S. Geological Survey (USGS) Earth Resources Observation and Science (EROS) Center through Earth Explorer (https://earthexplorer.usgs.gov/, accessed on 1 August 2021) for the month of October 2019 and used to calculate these indices. We chose October because it is the driest month showing the least snow and cloud cover over the study area and because vegetation types showed large variation in their spectral responses [46]. Among all the satellite scenes of October, we used the scene of 26 October 2019 since it showed the lowest snow level and cloud cover in the study area. Landsat 8 surface reflectance Level-2 dataset contains 11 separate images at different spectrum bands ranging from visible to thermal infrared (0.43–12.5 μm) in tiff format. Only three single band images (band 2 = blue, band 4 = red and band 5 = near infrared) were extracted and exported to ArcGIS 10.2 for calculating the NDVI and EVI for the study area. Landsat 8 surface reflectance-derived NDVI was calculated as a ratio between the red (band 4) and near infrared (band 5) values. Similarly, EVI was also calculated using band ratioing, but it takes into consideration some atmospheric and canopy background noise corrections. It includes correction value for canopy background, coefficients for atmospheric aerosol resistance for band 5 and band 2. These coefficients and correction values reduce background noise, atmospheric noise, and soil reflectance [47].

EVI is a more effective vegetation index developed to augment the vegetation signal with improved sensitivity in high above ground biomass areas and improved vegetation monitoring through a de-coupling of the canopy background signal and a reduction in atmospheric influences [48]. Furthermore, we used VIIRS 375 m standard Active and Thermal Anomalies satellite product (VNP14MGTML) processed by the University of Maryland and distributed by FIRMS to calculate the total number of active small-scale fires in the study area around NNS and ANS for the year 2019. This dataset will help in identifying the potential hotspots where nomadic people cleared the forest patches by burning the live tree to fetch wood for their energy need and construction of settlements.

### 2.4. Data Analyses

#### 2.4.1. Plant Traits

To evaluate if there is a variation of live and dead plants traits between sites, we conducted a Generalized Linear Model (GLM) with Gaussian error distribution followed by Likelihood Ratio test (LRT) using the "stats" and "car" [49] packages, respectively. The plant parameters analysed were DBH, height, growing stock volume density (GSVD), above ground biomass density (AGBD), below ground biomass density (BGBD), total biomass density (TBD), total carbon density (TCD), basal area, and density. Analyses were conducted in R software 4.0.0 [50]. Packages used in R software are mentioned ahead for each analysis.

#### 2.4.2. Diversity Indices, Fire Incidences, NDVI, and EVI between Sites

To compare the diversity indices (species richness, dominance, Shannon, Simpson, Pielou Evenness, Margalef Richness, equitability, and Fisher alpha) between sites, we conducted a GLM with Gaussian error distribution, except for species richness, in which we used Poisson distribution. Diversity indices were analysed only for living plants. To compare the number of fire incidences between sites, we conducted a GLM with Poisson error distribution. For NDVI and EVI, we also used a GLM, but with Gaussian error distribution. The NDVI and EVI were used as a proxy of forest cover density. More the NDVI and EVI more would be the canopy density of forests which indicated less disturbed forests by nomadic population and vice versa. The importance value index was used to assess which species dominate in the study area in terms of frequency of occurrence of

sampled species, the number of individuals per unit sampled area and the total basal area of sampled species. Various diversity indices assessed in this study can be categorised into two types. The first category represents indices that are used for defining the number or type of species present in the study area, whereas the second category of indices is used for assessing the dominance or evenness of the species in the study area. The species richness, evenness, and dominance often vary for a community with different sampling intensities. For example, the number of species for a small-sampled area may be few compared to the enumeration done for a larger area. Thus, the species richness value is often guided by sampling efforts. Greater the effort more could be index value. Hence, comparing indices with different sampling efforts can be misleading. Therefore, various indices were collected to visualise all such differences. Species richness was calculated as simply the total number of different species in the sample, whereas to compensate for the sampling efforts Margalef Richness Index was calculated as $(S - 1)/\ln(n)$, where S is the number of species in the sample, and $\ln(n)$ is the natural logarithm of the number of organisms collected. Dominance was calculated as 1-Simpson index that ranges from 0 to 1, where 0 means all species are equally present and 1 means one species dominates the community completely. Shannon and Simpson's indices were used to represent species diversity where Shannon accounts for the evenness of species distribution and Simpson accounts for the dominance of species. The evenness component was enumerated using Pielou Evenness, equitability, and Fisher alpha. Pielou Evenness was calculated as $H'/\ln(S)$, where H' is Shannon diversity and S is the total number of species in a sample. Equitability was calculated as Shannon diversity divided by the logarithm of the number of species. This helped to measure the evenness with which individuals are divided among the species present. Fisher alpha was calculated by the formula $S = a*\ln(1 + n/a)$ where S is the number of species, n is the number of individuals and a is the Fisher's alpha.

### 2.4.3. Plant Species Composition

To compare whether there is a difference in species composition between the two sampling sites (ANS and NNS), we used an NMDS followed by a Permutational Multivariate Analysis of Variance (PERMANOVA) with Euclidean distance and 999 permutations in the "RVAideMemoire" package [51]. To observe the contribution of each plant species to overall dissimilarities, we used the *simper*() function, which is a similarity percentage analysis based on the decomposition of Bray–Curtis dissimilarity, from package "vegan" [52]. Species richness was simply taken as a count of the total number of species in that particular plot. The main diversity indexes Shannon–Wiener, Simpson, Margalef richness index, Evenness Index, and Dominance index were calculated using Past software version 3.14 [53].

### 3. Results

#### 3.1. Plant Traits

Considering live plants, we found no significant difference between forest types for eight of the nine plant-related parameters evaluated (Figure 3; Table 1). We only found a significant difference for density (Figure 3i), in which ANS had a higher average density than NNS (Figure 3i; Table 1). On the other hand, in the case of dead plants, we found a significant difference in all nine plant-related parameters evaluated between forest types (Figure 4; Table 1). NNS had a higher value of all parameters evaluated (Figure 4; Table 1), except for height, which was higher in ANS forest type (Figure 4b; Table 1).

**Table 1.** GLM analyses of the plant-related parameters evaluated for live and dead plants between forests near and away to nomadic settlements in the Bangus Valley.

| | Live Plants | | | | Dead Plants | | | |
|---|---|---|---|---|---|---|---|---|
| | LR Test | *p*-Value | ANS (Mean ± SD) | NNS (Mean ± SD) | LR Test | *p*-Value | ANS (Mean ± SD) | NNS (Mean ± SD) |
| DBH (cm) | 0.583 | 0.4449 | 257.73 ± 42.63 | 289.32 ± 70.88 | 39.807 | **0.0460** | 241.66 ± 37.85 | 280.70 ± 9.91 |
| Height (meter) | 0.063 | 0.8012 | 17.84 ± 2.08 | 18.42 ± 4.12 | 41.574 | **0.0414** | 19.50 ± 1.50 | 17.73 ± 0.86 |
| GSVD (m$^2$ha$^{-1}$) | 0.596 | 0.4398 | 21.81 ± 8.58 | 17.05 ± 8.82 | 27.205 | **0.0001** | 4.04 ± 1.11 | 8.70 ± 1.40 |
| AGBD (Mgha$^{-1}$) | 0.963 | 0.3262 | 118.14 ± 43.96 | 89.17 ± 39.37 | 34.28 | **0.0001** | 21.85 ± 5.43 | 47.38 ± 6.82 |
| BGBD (Mgha$^{-1}$) | 1.266 | 0.2605 | 42.25 ± 15.09 | 31.23 ± 12.47 | 39.685 | **0.0001** | 7.80 ± 1.80 | 7.80 ± 2.27 |
| TBD (Mgha$^{-1}$) | 1.038 | 0.3083 | 160.39 ± 59.03 | 120.40 ± 51.74 | 35.641 | **0.0001** | 29.66 ± 7.23 | 64.33 ± 9.08 |
| TCD (Mg C ha$^{-1}$) | 1.038 | 0.3083 | 73.78 ± 27.15 | 55.38 ± 23.80 | 35.641 | **0.0001** | 13.64 ± 3.32 | 29.59 ± 4.18 |
| Basal area (m$^2$ha$^{-1}$) | 0.892 | 0.3449 | 92.02 ± 31.36 | 71.63 ± 29.64 | 42.599 | **0.0001** | 15.24 ± 4.44 | 36.54 ± 4.77 |
| Density (Nha$^{-1}$) | 4.235 | **0.0395** | 165.00 ± 38.72 | 105.0 ± 43.58 | 50 | **0.0001** | 32.5 ± 5.0 | 57.5 ± 5.0 |

Note: LR, Likelihood Ratio test; ANS, Away from nomadic settlements; NNS, near nomadic settlements. (DBH, Diameter at breast height; GSVD, Growing stock volume density; AGBD, Above ground biomass; BGBD, Below-ground biomass; TBD, Total biomass density; TCD, Total carbon density) Significant values are in bold.

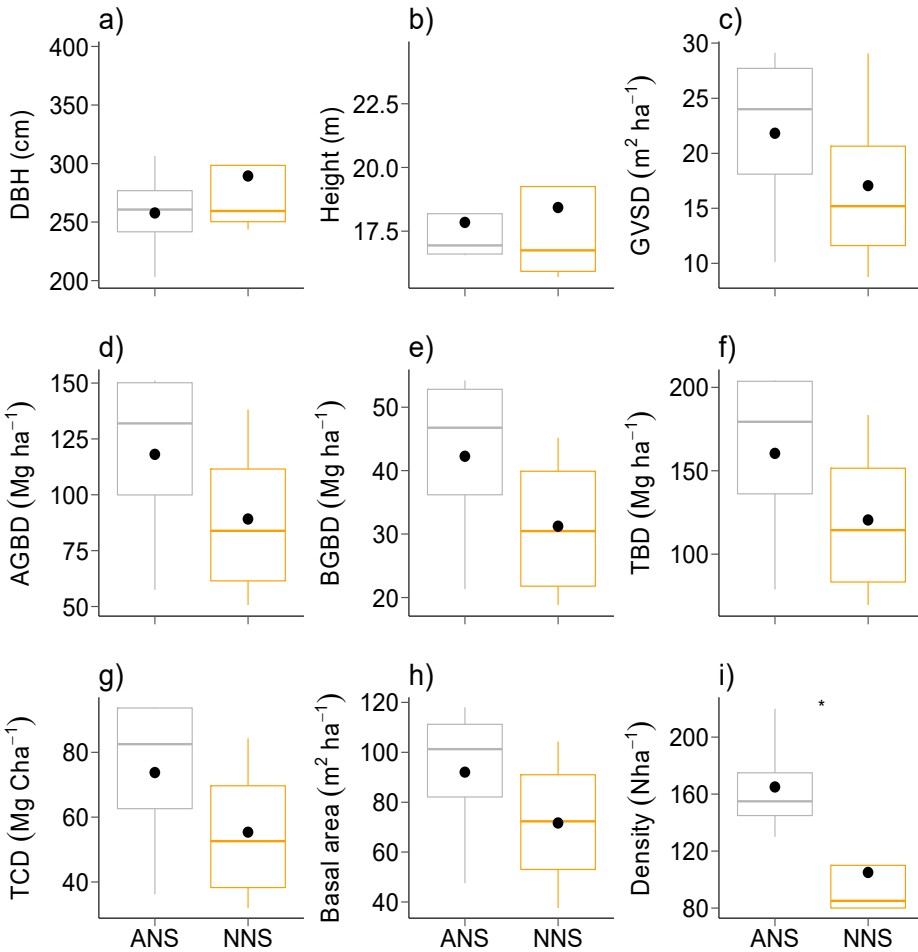

**Figure 3.** Variation of nine different plant traits of live plants between the forest types at Bangus Valley. (**a**) DBH, Diameter at breast height; (**b**) Height; (**c**) GSVD, Growing stock volume density; (**d**) AGBD, Above ground biomass; (**e**) BGBD, Below-ground biomass; (**f**) TBD, Total biomass density; (**g**) TCD, Total carbon density; (**h**) Bassal area; (**i**) Density. ANS, Away from nomadic settlements; NNS, near nomadic settlements. GLM results are depicted in Table 1. Black dots represent the mean. * $p < 0.05$.

### 3.2. Diversity Indices, fiRe Incidence, NDVI, and EVI

We did not find differences in species richness, dominance, Shannon, Simpson, and equitability indices between sites (Figure 5; Table 2). On the other hand, we found a significant difference of evenness, Margalef Richness Index, and Fisher alpha between forest types (Figure 5; Table 2). ANS had higher values of Margalef and Fisher, but lower values of evenness (Table 2). We found a significant difference in fire incidences, NDVI, and EVI between forest types (Figure 6; Table 2). NNS had higher values of these all parameters (Table 2).

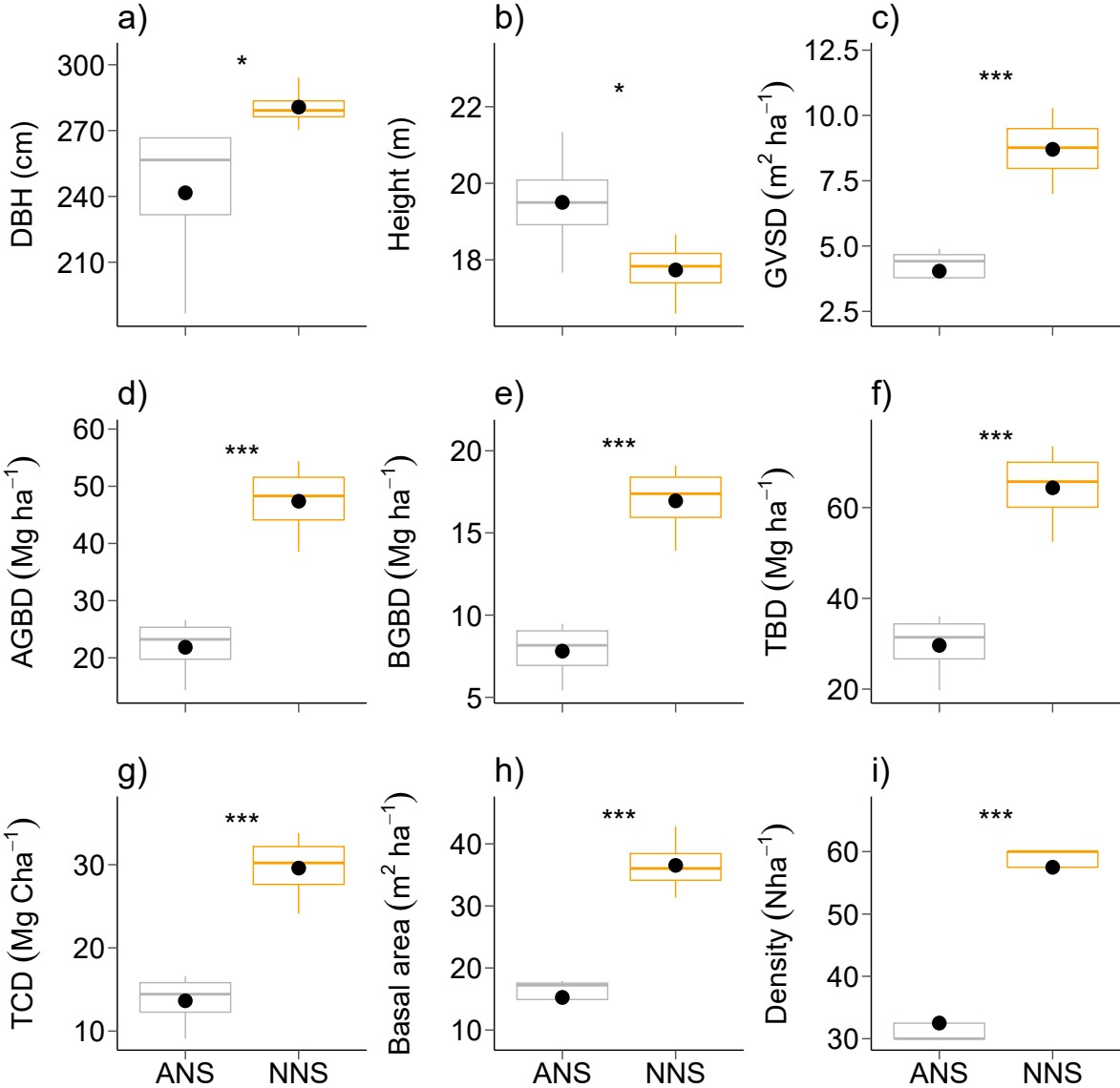

**Figure 4.** Variation of nine different plant traits of dead plants between the forest types at Bangus Valley. (**a**) DBH, Diameter at breast height; (**b**) Height; (**c**) GSVD, Growing stock volume density; (**d**) AGBD, Above ground biomass; (**e**) BGBD, Below-ground biomass; (**f**) TBD, Total biomass density; (**g**) TCD, Total carbon density; (**h**) Basal area; (**i**) Density. ANS, Away from nomadic settlements; NNS, near nomadic settlements. GLM results are depicted in Table 1. Black dots represent the mean. * $p < 0.05$, *** $p < 0.001$.

**Table 2.** GLM analyses of diversity indices and fire incidences, NDVI, and EVI between forest types.

| Indices | LR Test | *p*-Value | ANS (Mean ± SD) | NNS (Mean ± SD) |
|---|---|---|---|---|
| Species richness | 2.697 | 0.1005 | 29.75 ± 3.59 | 23.75 ± 2.36 |
| Dominance | 0.280 | 0.5966 | 0.09 ± 0.02 | 0.09 ± 0.01 |
| Shannon | 1.310 | 0.2524 | 2.81 ± 2.70 | 0.16 ± 0.09 |
| Simpson | 0.281 | 0.5957 | 0.90 ± 0.02 | 0.90 ± 0.14 |
| Evenness | 4.746 | **0.0293** | 0.56 ± 0.02 | 0.63 ± 0.05 |
| Margalef | 10.933 | **0.0001** | 6.68 ± 0.66 | 5.37 ± 0.42 |
| Equitability | 2.088 | 0.1485 | 0.83 ± 0.01 | 0.85 ± 0.02 |
| Fisher | 12.582 | **0.0001** | 18.62 ± 2.65 | 12.97 ± 1.75 |
| Fire incidences | 10.818 | **0.0010** | 0.75 ± 0.95 | 4.25 ± 1.89 |
| NDVI | 20.857 | **0.0001** | 0.23 ± 0.01 | 0.13 ± 0.04 |
| EVI | 60.956 | **0.0001** | 0.22 ± 0.01 | 0.12 ± 0.02 |

Note: LR, Likelihood Ratio test; ANS, Away from nomadic settlements; NNS, near nomadic settlements. Significant values are in bold.

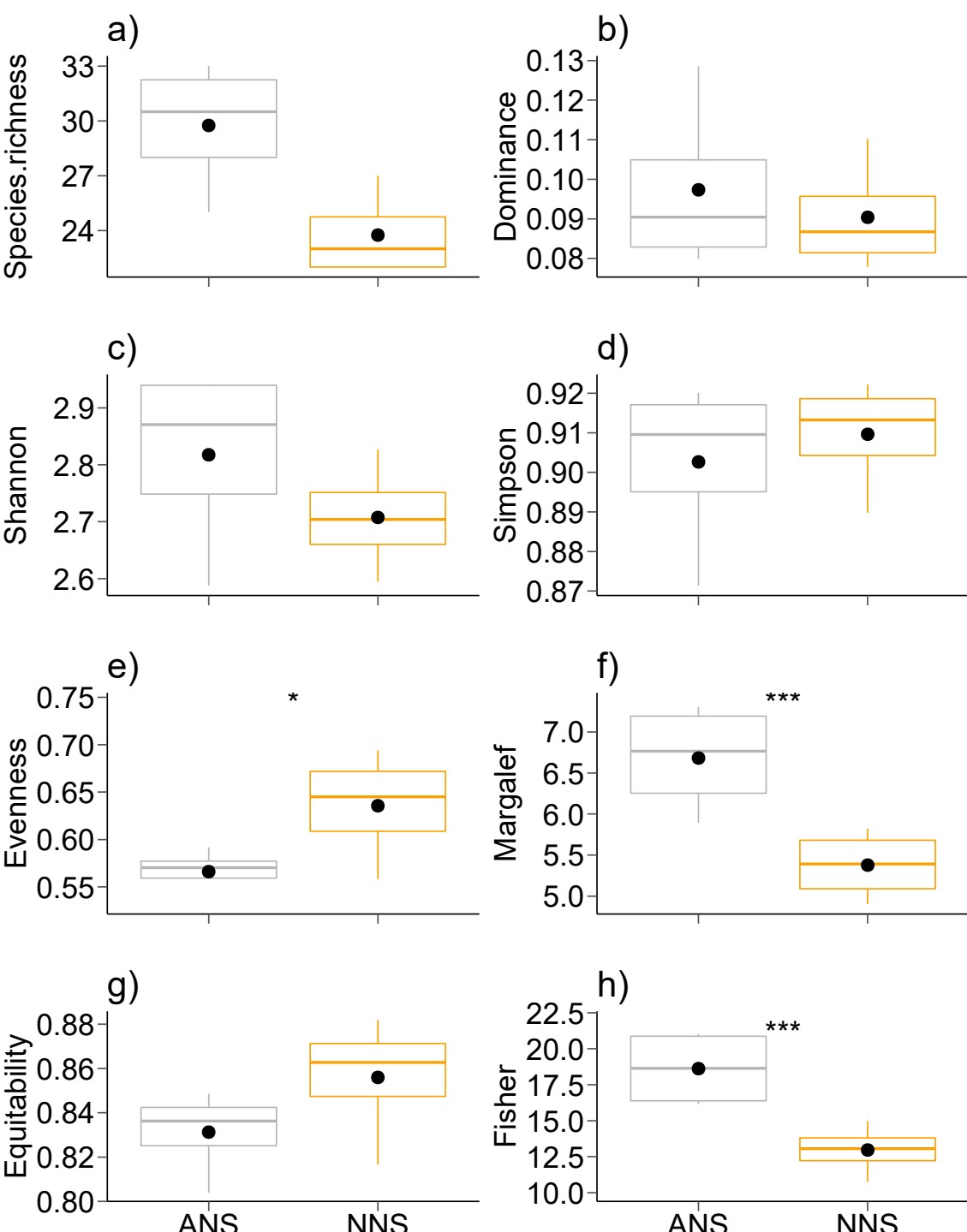

**Figure 5.** Variation of diversity indices between the forest types at Bangus Valley. (**a**) Species richness; (**b**) Dominance; (**c**) Shanon; (**d**) Simpson; (**e**) Evenness; (**f**) Margalef; (**g**) Equitability; (**h**) Fisher. ANS, Away from nomadic settlements; NNS, near nomadic settlements. GLM results are depicted in Table 3. Black dots represent the mean. * $p < 0.05$; *** $p < 0.001$.

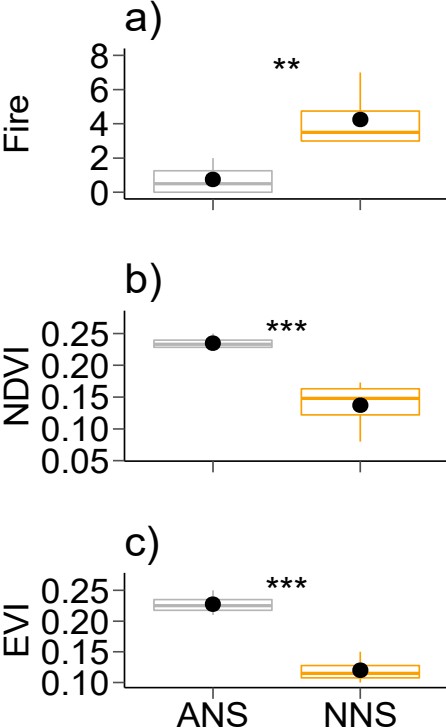

**Figure 6.** Variation of fire incidence, NDVI, and EVI between forest types. (**a**) Fire; (**b**) NDVI; (**c**) EVI. ANS, Away from nomadic settlements; NNS, Near nomadic settlements. GLM results are depicted in Table 2. Black dots represent the mean. ** $p < 0.01$, *** $p < 0.001$.

### 3.3. Variation of Plant Species Composition between Plant Communities

We found a significant variation in plant species composition between sites (Table 3). Out of 57 species found, three species contributed equal or more than 7.9% to the variation in plant species composition between sites, namely, *Sibbaldia cuneata* (9.2% contribution), *Poa annua* (7.9%), and *Abies pindrow* (7.9%), making up 25% of cumulative variation (Supplementary Material Table S1).

**Table 3.** PERMANOVA results comparing species composition between the four communities found in Fir Forest.

|  | Df | Sums Of Sqs | Mean Sqs | F | $R^2$ | Pr (>F) |
|---|---|---|---|---|---|---|
| Site | 1 | 370.44 | 370.44 | 3.350 | 0.3583 | 0.024 |
| Residuals | 6 | 663.44 | 110.57 |  | 0.6417 |  |
| Total | 7 | 1033.89 |  |  | 1 |  |

Note: This analysis was made with Euclidean distance and 999 permutations. Pairwise comparisons between communities are depicted in Table 3.

### 3.4. Satellite-Based Vegetation Indices and Small-Scale Forest Fires

In NDVI map, much of the pastureland and forest cover showed similar NDVI values, due to strong reflectance in near infrared wavelength from the lush green grass in the pasture (Figure 7). This interference is quite negligible in EVI, where the values of pastures are quite lower than the forest cover (Figure 7). The average NDVI and EVI values in a 3×3 pixels array around the center point of all the observed quadrants in NNS and ANS are shown in the bar charts in Figure 6. From the results of vegetation indices, it was observed that the average NDVI values in NNS and ANS were (0.14 ± 0.04) and (0.23 ± 0.03), respectively. The range of NDVI values in NNS and ANS were found to be (0.07–0.21) and (0.15–0.28), respectively. Similarly, the average EVI value in NNS and ANS was observed as (0.12 ± 0.04) and (0.23 ± 0.02), ranging (0.05–0.18) and (0.18–0.27), respectively. The results revealed that the average NDVI and EVI values around ANS were 1.64 and 1.91 times higher than values around NNS. This means that EVI had differentiated the disturbed and

undisturbed forest patch. Moreover, in an attempt to analyse the yearlong active small-scale open fires at NNS and ANS, the satellite data revealed that, despite the constraints that small-scale fires are difficult to capture by the VIIRS sensor, still around 40 fire episodes with >80% confidence quality flag were captured in the Bungus Valley during year 2019. The distribution of this fire site in and around NNS and ANS are shown in Figure 8. Out of these 40 fires episodes, 17 were observed in NNS and only 3 episodes in ANS, while the remaining 20 episodes were observed in other places in the study area.

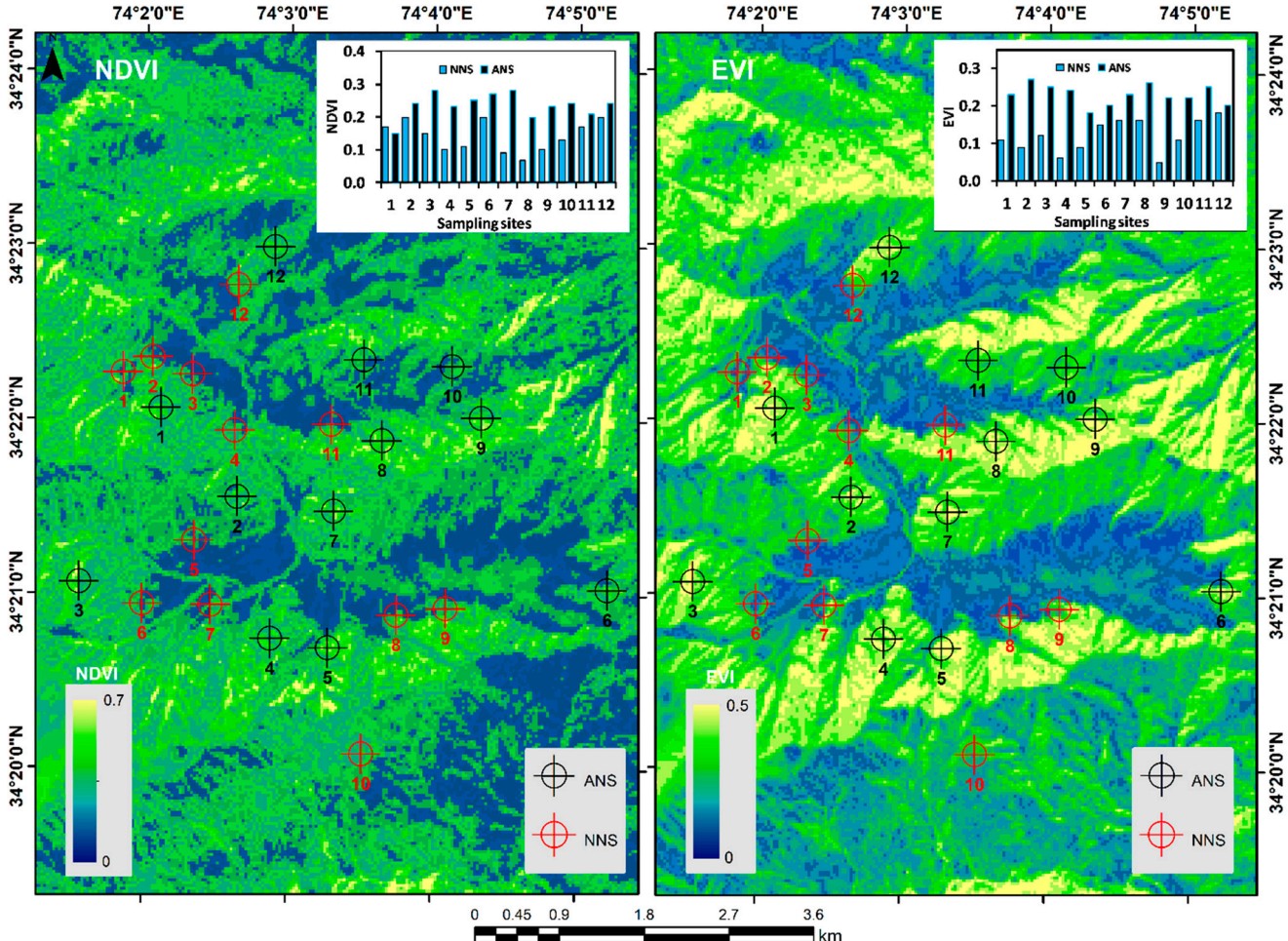

**Figure 7.** NDVI and EVI maps of Bangus Valley generated from Landsat 8 satellite imagery (26 October 2019) (ANS, Away Nomadic Settlement sites, NNS, Near Nomadic Settlement sites).

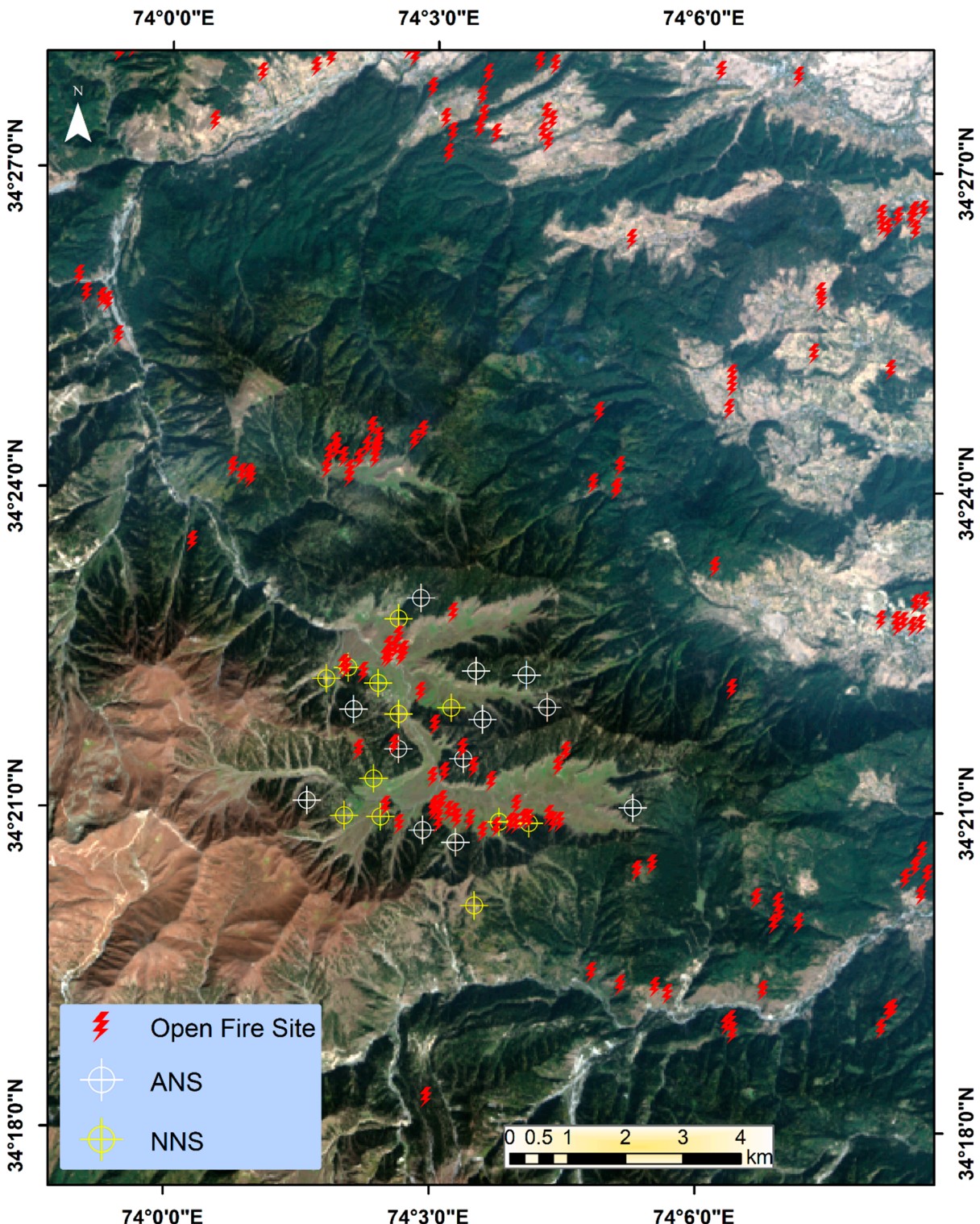

**Figure 8.** Distribution of small-scale forest fires in and around Bangus Valley.

## 4. Discussion

To ensure the long-term functioning and providing of ecosystem services, high altitude forests must be resilient to natural and anthropogenic disturbances. The capacity for climate change mitigation is determined by biomass recovery, which may be linked to the restoration of other forest qualities such as species composition, diversity, and structural attributes [42]. The degradation of forests and destruction of natural forest habitat due to

anthropogenic stresses significantly have an adverse effect on the Himalayan Forest ecosystems [54,55], particularly Kashmir Himalayas [36]. However, the frequent and intense disturbances (e.g., nomadic settlement, stem cutting, tree lopping, livestock grazing, degradation, and fire) may involve the cumulative effect of multiple stress and strongly affect patterns of biodiversity [56] and the structural attributes of Himalayan forests [43,57]. In this study, we investigated the impact of nomadic settlement and associated disturbances in high-altitude forests of Kashmir Himalaya, India. Here, we showed the following: (i) Plant community composition and diversity varied as a function of location and disturbance levels. (ii) The results revealed that the diversity parameters of plant species reflected the differences in the anthropogenic factors. (iii) We also observed that ecosystem properties (biomass and carbon stock) of the forest types decrease with the disturbance intensity due to nomadic settlements; furthermore, (iv) we also observed that dead plant parameters were significantly higher at near nomadic settlement forest types. Finally, (v) we found difference of fire incidences, NDVI, and EVI between forest types. Overall, our results fill some existing gaps of knowledge about impact of nomadic settlements on the variation in the species composition, diversity, and ecosystem properties in the high-altitude forest of Kashmir Himalaya.

The negative effect of disturbance on the species richness and total carbon stock is likely associated strong reduction in forest types near nomadic settlements. Human-induced disturbance culminates in habitat destruction, which results in the decrease of species richness with an increased degree of disturbance in near nomadic settlements in forests communities [43,58]. The loss of species richness exceeding 12% at the NNS forest type likely impair the contribution of biodiversity to ecosystem services and functioning, and thus to the livelihood of local users [59]. Hence, in this current study, on comparative analysis, the lesser values have been found for Shannon Index (H), Fisher alpha (FA), Margalef richness index (MRI), in the case of the community structure of forests for Near Nomadic settlements. Similar findings are reported by several previous studies in the Himalayas [43,60–62]. We found a significant difference in all nine plant-related parameters evaluated for the dead plants between forest sites. NNS had a higher value of all parameters evaluated, except for height, which was higher in ANS sites. The possible reason maybe that the nomadic people use the forest as the source of energy, timber for the construction of shelter and other livelihood requirements. Indicators of forest productivity (biomass) showed positive trends with the distance away from the nomadic settlements. In our study, the highest values (Mgha$^{-1}$) of AGBD, BGBD, and TBD were obtained for ANS forest type. Previous works also report similar trend in biomass parameters in Himalayan forests [63–65]. NNS forest type showed 2.1 times more average carbon loss (29.59 $\pm$ 4.18 MgCha$^{-1}$; mean $\pm$ SD) than ANS forest type (13.64 $\pm$ 3.32 MgCha$^{-1}$). Our results are in agreement with previous studies in Himalayan forests and elsewhere in the world. For example, [66] reported carbon loss of 25.5 MgCha$^{-1}$ from the forest of central Amazonia. Ref. [67] found that deforestation in human-occupied regions accounts for about 20% of the total C loss in Amazonia. Ref. [68] reported that about 38% of the forest area was cleared prior to settlement creation. Ref. [69] reported an average carbon loss of 31.33 MgCha$^{-1}$ from the forest of Pakistan Himalayas.

In the case of live trees, a significant difference was observed only for the "tree density" when comparing forests near and away from nomadic settlement sites. No significant difference in other remaining eight plant related parameters was found because nomadic population usually cut young trees with lesser diameter for the fodder and fuel, whereas they extensively extract dead trees for the fuel. As a result, for the live trees, the number of trees per ha (i.e., tree density) differed significantly between the forest sites near and away from nomadic settlements sites, whereas other parameters such as tree height, tree diameter, above and below ground biomass, total biomass, etc. attributed because of old mature trees did not have a significant difference. A significant difference in plant density, where ANS had a higher average density than NNS is because the nomadic settlements clear the forest for the livelihood needs and also to feed their cattle. They clear

forest around the settlements to create grazing fields. The density of stem cuttings is a prominent indicator of human-induced disturbance in forest ecosystem. The stem cutting density and basal area increase positively as the distance between forest and nomadic settlement decreases. A similar trend was observed by several workers from other parts of the Himalayas [43,59]. However, live stem density and basal area show an inverse trend with respect to the distance between forest and nomadic settlement. NNS forest type had the least stem density and basal area as compared to ANS forest type. Similar relatively higher values were reported at less disturbed sites by various workers in other parts of the Himalayas, revealing the negative impact of anthropogenic stresses on the forest structural attributes [65,70,71]. Due to the limited livelihood options, the nomadic people living in the vicinity of the high-altitude fir forests depend heavily upon coniferous tree species for timber, fuelwood, and leaf litter. The forest cutting near human settlements relies on the demand of settlers and this is especially true when the areas involved were previously dominated by large livestock herders [72]. In the present study, higher (35%) stem cutting was reported at NNS compared to ANS forest sites (16%), respectively. Fearnside [73] found that the human settlements near forest accounted for 30.5% of the deforestation activity in the Brazilian Amazon.

On the other hand, we discovered an ample open space near the nomadic settlement, increasing the abundance of weedy forbs and decreasing the species richness at these sites. Thus, forest sites away from nomadic had higher values of evenness. The Nibble impact, which causes massive grazing pressure, is reflected in the increase in scrub area in these nomadic settlement hotspots. Forests are turned to scrub area grasslands to accommodate secondary vegetation. Endless grazing may also cause sudden spikes in naked rocks because it destroys so much vegetative protection, speeding up the solifluction process and increasing the area under rocks and stones. Forest stem cuttings can significantly alter edaphic and microclimate conditions and shape the diversity and composition of the herbaceous layer [74,75]. Besides this, the intensity of canopy removal (amount of basal area removed or gap size) and degree of ground disturbance are other factors in altered the species mosaic in this part of the forest and favored the abundant growth of weedy flora, such as *Urtica dioica* L., *Daucus carota* L., *Capsella bursa-pastoris* (L.) Medik., *Cirsium falconeri* (Hook.f.) Petr., *Veronica laxa* Benth., *Ranunculus hirtellus* Royle, *Malva neglecta* Wallr., *Polygonum aviculare* L., *Erodium cicutarium* (L.), and *Mentha longifolia* L. around nomadic settlements, result in alterations of plant community composition [76]. It has been investigated that the high nutrient levels due to the increase in cattle faeces and urine may allow the establishment of weedy plants [77,78]. The unintentional introduction of weedy plant species to new habitats by creating disturbed habitats such as piospheres has created new opportunities for the dispersal of weedy plants in natural forest habitat [79,80]. The low species richness at the near nomadic forest sites may affect the livelihood of the nomadic people due to the high utilisation pressure of plant resources [81].

We found a significant difference in fire incidences, NDVI, and EVI between sites, in which NNS had higher values of all three parameters. The forest sites near nomadic settlements are prone to fire as the nomadic settlers sometimes intentional burn the forest for increasing grazing lands. In order to estimate biomass or carbon mapping, investigators use both field as well as Landsat data [82]. The results revealed that the average NDVI and EVI values around ANS were 1.64 and 1.91 times higher than values around NNS. Ref. [83] observed various NDVI values for closed and open strata for temperate forests of Western Himalayan regions where NDVI values for Himalayan Cedar was in the range of 0.32–0.17, while for Fir Spruce, it was 0.24–0.32 and 0.17–0.24 for Blue Pine, respectively. In a similar study by [46] in Lidder valley, NDVI is consistently decreasing when an undisturbed forest cover is degrading. It was also observed that NDVI decreases by 50% when the undisturbed forest cover is cleared for agricultural purposes and by about 33% when the dense forest cover was converted into degraded forest. Out of 40 fire episodes, 17 were observed in NNS and only 3 episodes in ANS, while the remaining 20 episodes were observed in other places in the study area. This dataset revealed that the nomadic population is disturbing

the surrounding biodiversity, and one of the most significant factors is the burning of live trees around their settlements. Apart from near nomadic population sites, the dataset also identifies some other hotspots in the valley where several consistent fire episodes were observed. Therefore, this dataset can be an important input parameter for delineating the highly disturbed areas in the study area.

## 5. Conclusions

This study is one of the pioneering works to estimate the level of disturbance caused by the nomadic settlements in the Himalayan forests. It was observed that the three species contributed equal to or more than 7.9% to the variation in plant species composition between sites. The live stem density and basal area show an inverse trend with respect to the distance between forest and nomadic settlement. We found a significant difference in all nine plant-related parameters evaluated for the dead plants between forest sites. We reported higher stem cutting at NNS (35%) as compared to ANS forest sites (16%). With the intense human presence and increasing grazing intensity, the understory forest vegetation is converted into scrub areas. In addition, the increasing grazing activity at NNS results in the exposure of bare rocks. The study also provides insight into the need for additional policies to reduce forest disturbance by examining the relative significance of nomadic settlement on forests. However, local engagement during all of the planning stages should be brought into consideration, and the social ecology approach should be followed while dealing with the indigenous communities. Forests that have been disturbed by the nomadic tribes need to be restored with adequate measures that ensures livelihood of nomadic people outside these forest lands. Development of grazing lands outside the forested lands to support cattle population must be a top-level priority. Educating tribal population to bring them into mainstream for other livelihood options is need of the hour. The permanent huts of nomadic tribes inside the forest lands should be removed in consultation with the tribal community after providing alternative livelihood options that do not require grazing of cattle inside forests. In this regard, the forest administration must expand outreach and educational initiatives regarding the importance of forests, and alternative livelihood opportunities should be developed while dealing with the nomadic pastoralists. This study evaluates influence of nomadic settlements on the high-altitude forest, more importantly, the corban stock potential in Himalayan region, which will facilitate better climate change mitigation options together with benefits of the ecosystem through forest recovery of disturbed areas having nomadic influence.

**Supplementary Materials:** The following are available online at https://www.mdpi.com/article/10.3390/su132212497/s1, Table S1. Contrast table of the contribution of individual plant species to the overall Bray–Curtis dissimilarity of species composition between sites (ANS and NNS). Av. dis., Average dissimilarity; SD, Standard deviation; Av Com1, Average Community 1; Av Com2, Average community 2; Cum., Cumulative; Cont., Contribution.

**Author Contributions:** S.M.H. conceived the research idea. S.M.H., U.Y. and F.A.L. collected field data. S.M.H., U.Y., E.S.C., M.K. and I.U.R. analysed data, obtained results, wrote the original draft of the manuscript, and revised it at multiple stages. A.H., E.F.A., M.A.A., A.A.A., M.A., M.A.K., F.A.L., U.K. and F.I. helped in data analysis, review, and editing. All authors have read and agreed to the published version of the manuscript.

**Funding:** The authors would like to extend their sincere appreciation to the Researchers Supporting Project Number (RSP-2021/356), King Saud University, Riyadh, Saudi Arabia.

**Institutional Review Board Statement:** Not applicable.

**Informed Consent Statement:** Not applicable.

**Data Availability Statement:** Not applicable.

**Acknowledgments:** The authors are thankful to those who directly or indirectly helped them during the study. The authors would like to extend their sincere appreciation to the Researchers Supporting Project Number (RSP-2021/356), King Saud University, Riyadh, Saudi Arabia.

**Conflicts of Interest:** The authors declare no conflict of interest.

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
