# Peer review of "Long-Term Impact of Transhumance Pastoralism and Associated Disturbances in High-Altitude Forests of Indian Western Himalaya"

_sustainability, doi:10.3390/su132212497_

Round 1
Reviewer 1 Report
The authors conducted a study on the impact of nomadic settlements on plant species composition and ecosystem properties in Himalayan forests. They found a variation in plant species among study areas near versus away from a nomadic settlement. Forests near nomadic settlements were also more prone to forest fire incidence.
What I am missing is more details on the study design. Nomadic settlements are by definition temporary. Therefore it would be important to take into account how long the nomadic settlement existed before the study was conducted and likewise, whether any of the study areas defined as a ‘away nomadic settlement’ were in the past a ‘near nomadic settlement’. These factors influence the composition of vegetation. Another important aspect is whether the observed characteristics are stable in time. I suggest the authors consider including a temporal factor in their study design.
Specific comments:
L105-106: Consider rewording to make the sentence clearer.
L112: Why ‘aimless’? Cattle move with an aim – to find some plants to eat…
L112: The term ‘populace’ is usually used when talking about people, not animals.
L133-137: Please justify using each of the listed methods. Otherwise, it comes across as if you ran your data through any type of test and calculated any type of index you could think of.
L180-182: You measured a distance of 200-250 metres with a measuring tape? Would it not be easier to use satellite imagery and GIS software for this?
Discussion: Please discuss why you did not find significant differences between the forest types in eight of the nine plant-related parameters.
Author Response
#Response to Reviewer 1
Point 1:The authors conducted a study on the impact of nomadic settlements on plant species composition and ecosystem properties in Himalayan forests. They found a variation in plant species among study areas near versus away from a nomadic settlement. Forests near nomadic settlements were also more prone to forest fire incidence.
What I am missing is more details on the study design. Nomadic settlements are temporary. Therefore, it would be important to consider how long the nomadic settlement existed before the study was conducted and likewise, whether any of the study areas defined as a ‘away nomadic settlement’ were in the past a ‘near nomadic settlement’. These factors influence the composition of vegetation. Another important aspect is whether the observed characteristics are stable in time. I suggest the authors consider including a temporal factor in their study design.
Response 1:The reviewer has raised a very valid point that we missed narrating in the original draft. The suggestion is very much important and relevant. In the study area section, we have included additional information to describe temporal dimensions in the study to make the situation and facts stronger for which the present study was implemented.
Specific comments:
Point 2:L105-106: Consider rewording to make the sentence clearer.
Response 2:Earlier line “The nomadic settlements have influenced the forests of the Himalayas already for millennia [26,27]; however, the impact of these localized changes has become recently effective concerning the climate and land use changes [28].” Has now been replaced with “The nomadic settlements have influenced the forests of the Himalayas already for millennia [26,27]; however, the influence of climate change and land use changes have further aggravated the impact in recent decades[28].”
Point 3:L112: Why ‘aimless’? Cattle move with an aim – to find some plants to eat…
Response 3:Thanks for highlighting the issue. The earlier line “The aimless munching by a limitless dairy cattle populace around the nomadic settlements also leads to invasion by several weed plants” has now been revised as “The grazing pressure by a limitless dairy cattle population around the nomadic settlements also leads to invasion by several weed plants.”
Point 4:L112: The term ‘populace’ is usually used when talking about people, not animals.
Response 4:The word “populace” has been replaced with “population”
Point 5:L133-137: Please justify using each of the listed methods. Otherwise, it comes across as if you ran your data through any type of test and calculated any type of index you could think of.
Response 5:In fact, we used all listed indices and attributes. The justification for using each of them has been supplemented under the Methodology section.
Point 6:L180-182: You measured a distance of 200-250 meters with a measuring tape? Would it not be easier to use satellite imagery and GIS software for this?
Response 6:The measurements were taken in the field and thus measuring tape was used.
Point 7:Discussion: Please discuss why you did not find significant differences between the forest types in eight of the nine plant-related parameters.
Response 7:We observed a difference for the dead trees for all nine studied plant traits whereas for live trees only a significant difference in “tree density” was observed when comparing forests near and away from nomadic settlement sites. No significant difference in the remaining eight plant-related parameters was observed in the case of live tree attributes because nomadic populations usually cut young trees with lesser diameter for the fodder and fuel whereas they extensively extract dead trees for the fuel. As a result, for the live trees, the number of trees per ha (i.e., tree density) differed significantly between the forest sites near and away from nomadic settlements sites, whereas other parameters such as tree height, tree diameter, above and below ground biomass, total biomass, etc. attributed because of old mature trees did not have a significant difference. The same has now been indicated in the revised MS under the discussion section (3rdparagraph).

Reviewer 2 Report
The present article is useful for formulating policies in conserving the native vegetation.
Please add a table with distribution of land cover areas and change rate analysis.
Discussion section will benefit from presenting data on loss of forest area due to transhumance small fires per 10years trend at least
This article will benefit from a section on transhumant impact on physiography.
Conclusion section is lacking clarity on policy recommendations for the benefit of the ecosystem through forest recovery and reforestation programmes in line with livelihood of local people and transhumant activity.
Author Response
#Response to Reviewer 2
Point 1:The present article is useful for formulating policies in conserving the native vegetation.
Please add a table with distribution of land cover areas and change rate analysis.
Response 1:The study was limited for sites near to the selected nomadic settlement sites which otherwise has same definition of land use i.e., “Forest”. So, in actual there is no land use also change the land cover remains the same i.e., covered with the trees but with reduced tree density. So, we did not focus on land use land cover change which will require a different orientation of the paper. We would be thankful if the reviewers consider our viewpoint.
Point 2:Discussion section will benefit from presenting data on loss of forest area due to transhumance small fires per 10 years trend at least.
Response 2:Again, such results and discussion will require a different orientation to study the facts. We plan to develop a separate follow up paper exclusively to discuss forest fire related trends and the influence of the nomadic population. Whereas, in the present study we focus on the major dominant factors (including the number of fire incidences) that differ between forests near to nomadic settlements and away from the settlements.
Point 3:This article will benefit from a section on transhumant impact on physiography.
Response 3:This is an important suggestion and the major impacts on physiography have already been included in the study. Other physiographic aspects that usually remain constant and not affected by the nomadic population has not been covered.
Point 4:Conclusion section is lacking clarity on policy recommendations for the benefit of the ecosystem through forest recovery and reforestation programmes in line with livelihood of local people and transhumant activity.
Response 4:This is a very important suggestion. We have included policy related recommendations under the Conclusion section to make this section useful and focused.

Reviewer 3 Report
Needs a careful proofreading for typos: smapling/sampling; looping/lopping; Lansat/Landsat. Should "habitat alternation" be "habitat alteration"?
Author Response
#Response to Reviewer 3
Point 1:Needs a careful proofreading for typos: smapling/sampling; looping/lopping; Lansat/Landsat. Should "habitat alternation" be "habitat alteration"?
Response 1:The entire MS has now been double-checked to ensure omission of such typo errors and other grammatical mistakes.
Authors are thankful to anonymous reviewers for their valuable suggestions, comments and critical input that helped to improve the MS.

Round 2
Reviewer 1 Report
Dear authors,
thank you for your revisions. I still have to come to the point I raised previously:
Point 5:L133-137: Please justify using each of the listed methods. Otherwise, it comes across as if you ran your data through any type of test and calculated any type of index you could think of.
To which your response was:
Response 5:In fact, we used all listed indices and attributes. The justification for using each of them has been supplemented under the Methodology section.
Having checked the revised manuscript, I saw that the justification for using each index was in fact NOT added to the Methods section. Please include an explanation for why each index was used, i.e. why it made sense to use all of them.
Author Response
Response to Reviewer
Point 1:
L133-137: Please justify using each of the listed methods. Otherwise, it comes across as if you ran your data through any type of test and calculated any type of index you could think of.
To which your response was: Response 5: In fact, we used all listed indices and attributes. The justification for using each of them has been supplemented under the Methodology section.
Having checked the revised manuscript, I saw that the justification for using each index was in fact NOT added to the Methods section. Please include an explanation for why each index was used, i.e. why it made sense to use all of them.
Response 1:
The suggestion is valuable for the common reader not working in the domain of ecology whereas all the indices mentioned in the MS are famous one and mostly people are quite aware about its use. However, following the suggestion of reviewer and to make it further clear to the common readers, uses of various indices has further been elaborated under Methodology section. The diversity is often defined by three components of richness (total number of species) and the evenness of presence of species. Again, these are often guided by the sampling efforts and thus different indices were formed to compensate sampling efforts by accounting number of total plants collected. Various indices used in the study and its explanation and need to include in the analysis is discussed one by one below. All these explanations now have been added in the Methodology section 2.4.2 (can be seen in the MS file with changes marked in track change mode).
- Importance Value Index (IVI): this was used to assess which species dominates in the study area in terms of frequency of occurrence of sampled species, number of individuals per unit sampled area and the total basal area of sampled species.
- Normalised difference vegetation index (NDVI) and enhanced vegetation index (EVI): these indices were used as a proxy of forest cover density. More the NDVI and EVI more is the canopy density of forests means less disturbed forests by nomadic population and vice versa.
- Species richness: total number of different species present in the study area.
- Dominance: 1-Simpson index. Ranges from 0 (all species are equally present) to 1 (one species dominates the community completely).
- Shannon index: A diversity index, taking into account the number of individuals as well as number of species. Varies from 0 for communities with only a single species to high values for communities with many species, each with few individuals.
- Simpson index: 1-dominance. Measures 'evenness' of the community from 0 to 1.
- Margalef index: (S-1)/ln(n), where Sis the number of species in the sample, and n is the number of organisms collected.
- Pielou Evenness: H'/ln(S) where H' is Shannon diversity and S is the total number of species in a sample.
- Equitability: Shannon diversity divided by the logarithm of number of species. This measures the evenness with which individuals are divided among the species present.
- Fisher alpha: a diversity index, defined implicitly by the formula S=a*ln(1+n/a) where S is number of species, n is number of individuals and a is the Fisher's alpha.
